# An Overview of Pituitary Incidentalomas: Diagnosis, Clinical Features, and Management

**DOI:** 10.3390/cancers14174324

**Published:** 2022-09-03

**Authors:** Shigeyuki Tahara, Yujiro Hattori, Koji Suzuki, Eitaro Ishisaka, Shinichiro Teramoto, Akio Morita

**Affiliations:** 1Department of Neurological Surgery, Nippon Medical School, 1-1-5 Sendagi, Bunkyo-ku, Tokyo 113-8603, Japan; 2Department of Anatomy and Neurobiology, Graduate School of Medicine, Nippon Medical School, 1-1-5 Sendagi, Bunkyo-ku, Tokyo 113-8602, Japan; 3Department of Neurosurgery, Juntendo University School of Medicine, 3-1-3 Hongo, Bunkyo-ku, Tokyo 113-8431, Japan

**Keywords:** pituitary incidentaloma, pituitary neuroendocrine tumor, Rathke cleft cyst, imaging, diagnosis, management, surgery

## Abstract

**Simple Summary:**

A pituitary incidentaloma is a pituitary tumor or mass that is incidentally discovered in imaging studies which have been performed for reasons other than the symptoms of pituitary lesions. The majority of pituitary incidentalomas are pituitary neuroendocrine tumors (PitNETs) and Rathke cleft cysts. PitNETs have received attention because of their distinction from pituitary adenoma in the new World Health Organization (WHO) classification. The natural history of PitNETs is partially known, and the management of pituitary incidentalomas has been determined based on this history; however, the pathology of PitNETs has significantly changed with the new WHO classification, and studies with a high level of evidence are required to consider treatment guidelines for pituitary incidentalomas.

**Abstract:**

Pituitary incidentalomas are tumors or mass lesions of the pituitary gland. These are incidentally discovered during imaging studies for symptoms that are not causally related to pituitary diseases. The most common symptom that triggers an examination is headache, and the most common type of pituitary incidentalomas are pituitary neuroendocrine tumors (PitNETs) and Rathke cleft cysts. The existing treatment strategy is controversial; however, surgical resection is recommended in cases of clinically non-functioning PitNETs with optic chiasm compression. In contrast, cystic lesions, such as Rathke cleft cysts, should be followed if the patients are asymptomatic. In this case, MRI and pituitary function tests are recommended every six months to one year; if there is no change, the follow-up period should be extended. The natural history of PitNET is partially known, and the management of pituitary incidentalomas is determined by this history. However, the pathogenesis of PitNET has significantly changed with the new World Health Organization classification, and follow-up is important based on this new classification. Therefore, a high level of evidence-based research is needed to consider treatment guidelines for pituitary incidentalomas in the future.

## 1. Introduction

A pituitary incidentaloma is defined as a tumor or mass of the pituitary that is discovered incidentally on imaging studies, including computed tomography (CT) and magnetic resonance imaging (MRI), performed for reasons other than symptoms caused by pituitary lesions, such as chronic headache, neurological symptoms in the head and neck, dizziness, head trauma, or physical examination [1,2,3,4,5,6,7]. However, the definition of pituitary tumors or mass lesions varies between reports. Some reports include only pituitary neuroendocrine tumors (PitNET) among incidentally discovered pituitary tumors and masses in imaging findings [1,2]. However, there are few studies that include all tumors and mass lesions, including Rathke cleft cysts [3,4,5,6]. In addition, pituitary incidentalomas referred to here are identified in imaging studies performed for reasons other than the lesions that led to their discovery, but additional studies may reveal symptoms related to these tumors. These include abnormalities in pituitary function during endocrine testing, and visual field abnormalities during neuro-ophthalmological testing. Such patients are sometimes called symptomatic pituitary incidentalomas, as opposed to asymptomatic pituitary incidentalomas [8]. It is difficult to determine whether a headache is associated with the lesion. Headaches are common in pituitary disease, and are reported to be present in more than one third of PitNET patients [9]. In this article, we review the epidemiology, clinical and radiological features, natural history, follow-up, and treatment of pituitary incidentalomas to provide a benchmark for their clinical management.

The World Health Organization (WHO, Geneva, Switzerland) classification, revised in 2022, changed the term pituitary adenoma to pituitary neuroendocrine tumor (PitNET)/adenoma [10]. However, we prefer to use the term PitNET in this context. Therefore, we have unified the term PitNET in this review.

## 2. Frequency of Pituitary Incidentalomas

### 2.1. Frequency at Autopsy

There are numerous reports of PitNETs discovered incidentally during autopsy [11,12,13,14,15,16,17,18,19,20,21,22,23,24,25,26,27,28,29,30,31,32,33,34,35,36] (Table 1).

The earliest report was that of Costello et al., who found adenomas in 22.5% of 1000 autopsy cases [11]. In all reports, the frequency ranged from 1.5% to 31.1%. However, when examining many cases (>1000), Hardy et al. reported 2.7%, McCormick et al. 8.8%, Schwesinger et al. 9.5%, Teramoto et al. 5.1%, and Buurman et al. 11.0%, with few reports exceeding 10% [13,14,22,27,32]. The most recent report from the Iranian Forensic Medicine Organization found pituitary tumors in 61 of 485 cases (12.6%). Compared to cases without tumors, there were no significant differences in mean age, sex, or body mass index [36]. A meta-analysis by Ezzat et al. found an estimated prevalence of 14.4% in the autopsy cases of tumors [31]. As described above, there is considerable variation among reports. This may be due to the age and sex of the population, as well as the method used to prepare tissue sections, and the criteria used by the diagnostic physicians. Teramoto et al. found a kind of incidental lesion in 17.8% of pituitary autopsy cases. The frequency of incidental pathological lesions larger than 2 mm was considered clinically significant, as they were detectable using MRI or other imaging studies. They comprised 6.1% of the total cases, with PitNETs and hyperplasia accounting for 2% and Rathke cleft cysts for 3.7% [27]. Moreover, they suggested that incidental lesions should be considered as a cause of false-positive findings when imaging reveals functional pituitary microadenomas manifesting with Cushing’s disease. [27].

### 2.2. Frequency during Imaging

There are many reports on the frequency of pituitary incidentalomas found during imaging studies [21,37,38,39,40,41,42,43,44,45,46,47,48,49,50,51] (Table 2).

In an MRI study, Chong et al. performed T1-weighted non-contrast MRIs on 52 patients with 3 mm slices to search for micro PitNETs, and found 38.5% with findings suspicious for micro PitNETs [37]. Hall et al. performed contrast-enhanced pituitary MRIs on 100 healthy volunteers and found that 10% had findings corresponding to PitNETs [38]. Yue et al. studied many patients (3672). They found PitNETs in only 0.16% of cases [40]. Furthermore, a 1.5 Tesla non-contrast brain MRI study of 2000 patients in the general population found macro PitNETs in 6 of 2000 patients (0.3%) [41], and a 1.5 Tesla MRI study of middle-aged and older patients aged 50–66 years found pituitary tumors in only 0.3% of cases [47]. In a population of 700 older patients with a mean age of 72.5 years, there were only two cases (0.3%) of PitNETs [44]. In contrast, there is a report that examines pediatric cases under 18 years [50], wherein non-enhancement lesions in the pituitary gland were observed in 76 (20.8%) of 365 cases. Hegenscheid et al. reported the results of whole-body MRI in 2500 patients covering a wide age range of 21 to 88 years. They found nine cases (0.36%) of PitNETs and four cases (0.16%) of pituitary cysts [45]. A prospective study on the incidence of true incidental sellar lesions detected using CT and MRI in a large medical center found they were present in 45 of 3840 patients (1.2%), and significantly more common in hospitalized patients [52]. In addition, a recent report from Germany used three 0.8 mm isotropic three-dimensional imaging sequences with high-resolution 3 Tesla MRI. A meta-analysis by Ezzat et al. found an estimated prevalence of incident tumors in imaging studies of 22.5% [31]. The frequency in imaging studies, again, as in autopsy cases, may depend on the age of the population, sex, the expertise of the diagnosing physician, and most importantly, the imaging method. Particularly, the more detailed the imaging of the pituitary region using contrast media, the higher the detection rate, which should be evaluated with caution.

In a report on the quantification of pituitary metabolic activity on positron emission tomography (PET) scans [42], the files of 40,967 patients (20,220 men and 20,747 women) who underwent whole-body 18F-fluorodeoxyglucose (FDG) PET/CT for evaluating malignancy (n = 35,147), benign disease, or cancer screening (n = 5820) were retrospectively reviewed. A local increase in pituitary FDG uptake was observed in 30 patients (0.073%); among them, 94.7% had findings suggestive of pituitary mass on MRIs performed on 19 patients. Hoang et al. also concluded that if a pituitary lesion is incidentally detected by FDG PET/CT and the composition, size, and mass effect of the lesion cannot be evaluated, the patient may have limited life expectancy or severe complications, and MRI of the pituitary region should be performed to evaluate the lesion [49].

The detection rate of pituitary incidentalomas in brain screening examinations is less than 1%, whereas the detection rate in imaging studies performed for the purpose of detecting pituitary lesions has been reported to be more than 10%. As mentioned previously, the lesions that can be visualized in imaging studies are 2 mm or larger, and the frequency of pituitary incidentalomas in a large number of autopsy case reports focusing on lesions larger than 2 mm was 6.1%. Considering these results, the actual frequency of pituitary incidentalomas is expected to be 5–10%.

## 3. Opportunity to Detect Pituitary Incidentalomas

There are several reports on the detection of pituitary incidentalomas [5,6,8,53,54,55,56,57,58]. In a nationwide survey in Japan, headache was the most common cause for identification (37.5%), followed by medical examination (13.2%), close examination for other intracranial diseases (13%), dizziness (11.5%), and head injury (6.9%) [5]. A Canadian patient database reported headache in 28% of patients, dizziness in 12%, stroke in 9%, and 7% for head trauma [55]. In a prospective study of patients aged 18 years and older in the United States, head trauma was the most common cause (13%), followed by dizziness (12.2%), primary eye disorder (6.1%), and sinus disease (5.3%) [57]. In a report from a single center in Japan, headache was responsible for 31.6%, medical examination for 21.5%, dizziness for 17.7%, and head injury for 10.1% of pituitary incidentalomas for which surgical treatment was selected [58]. These results indicate that headache is the most common initial symptom. Furthermore, many cases reported in Japan were discovered during medical check-ups. This is likely due to the unique Japanese brain check-up system.

## 4. Types of Pituitary Incidentalomas

When a pituitary lesion is discovered incidentally, a detailed imaging study is often performed, such as contrast-enhanced pituitary MRI. Table 3 shows the typical neoplastic or mass lesions that occur in and around the pituitary gland.

These can be broadly divided into solid lesions and cystic lesions, with PitNETs being the most common in the former, and Rathke cleft cysts the most common in the latter [5,54,59]. In a nationwide survey in Japan, the most common presumptive diagnosis was clinically nonfunctioning PitNETs (64.0%), followed by Rathke cleft cysts (27.5%) [5]. An analysis of the Japanese reimbursement database showed a generally similar result, with the percentage of clinically nonfunctioning PitNETs at 66.9% [60]. In contrast, a report from a single institution in Japan showed that 73 of 139 patients (52.5%) had cystic lesions strongly suggestive of Rathke cleft cysts, and 66 patients (47.5%) had PitNETs [2]. Although these results are only a presumptive diagnosis, and the exact pathological diagnosis is unknown, because not all incidental pituitary tumors are operated on, 91% of sella turcica tumors that required surgery have been reported to be PitNETs [61]. The immunohistochemical findings of PitNETs that underwent surgery show that 20% were multihormone-producing, 15% were gonadotropin-positive, 10% were growth hormone (GH)-positive, and 50% were negative for all anterior pituitary hormones [6]. However, this study was based on the previous WHO classification, 4th edition, which was revised in 2017. A study based on the current transcription factor-based classification of PitNETs found that among clinically nonfunctioning PitNETs, 74.7% were gonadotroph PitNETs, 10.1% were corticotroph PitNETs, 5.1% were somatotroph PitNETs, and 8.1% were null cell PitNETs [58]. Therefore, gonadotroph PitNETs were the most common, a finding consistent with the pathological findings of clinically nonfunctioning PitNETs, including symptomatic tumors [62]. The prevalence of PitNETs has also been reported in several countries, mainly in Europe [63,64,65,66,67,68,69]. Among these, reports from Iceland, Belgium, Switzerland, and the United Kingdom indicate a high prevalence of PitNETs, ranging from 77.6 to 115.57 per 100,000 population [63,64,65,66], with Iceland reporting a yearly increase. However, the prevalence in Finland, Sweden, and Argentina were lower than those from the countries mentioned above, ranging from 3.9 to 7.39 per 100,000 population [67,68,69]. In contrast, Japan reported prevalence rates for acromegaly, with a prevalence rate of 9.2 per 100,000 population in 2015–2017 [70].

When a pituitary incidentaloma is detected, it is important to differentiate between neoplastic and non-neoplastic lesions. In particular, we need to be careful to distinguish a primary empty sella (PES) from other cystic lesions. PES was four times more common in women, and 79% of cases had incidental findings on MRI. In addition, 28% of patients with PES had some form of hypopituitarism [71]. It has also been reported that 40.5% of patients with PES had hypopituitarism, of which 29% of PES were found to have an abnormal pituitary function incidentally [72]. In addition, pituitary hyperplasia should be noted. In particular, pituitary hyperplasia due to primary hypothyroidism is observed as a marked contrast with equal T1 and slightly longer T2 signals on MRI [73].

## 5. Evaluation at the Time of Discovery and Natural History

The American Endocrine Society published well-known guidelines for the clinical management of pituitary incidentalomas in 2011 [7]. The French Society for Endocrinology has also reported on the management of clinical PitNETs among pituitary incidentalomas [74]. Molitch also proposed an algorithm for the follow-up of PitNETs among pituitary incidentalomas [75,76,77,78]. However, expert opinions from Brazil [79] and treatment guidelines from neuroradiologists and endocrinologists can also be found [80,81]. Hitzeman et al. also provided guidelines for the initial management of incidental tumors of eight organs, stating that patients presenting with pituitary incidentalomas should undergo pituitary-specific magnetic resonance imaging if the lesion is 1 cm or larger, or if it abuts the optic chiasm [82].

### 5.1. Evaluation at the Time of Detection

When pituitary incidentalomas are first detected, a rigorous medical history exploration and physical examination are recommended [7].

Giraldi et al. conducted a single-center study of surgically treated acromegaly over a 22 years and found that 17% of the cases were identified as pituitary incidentalomas and approximately half of these patients presented otorhinolaryngological symptoms [83]. This indicates the importance of collaboration with other departments.

However, it is also important not to overlook the signs of hypopituitarism. The frequency of hypopituitarism in incidentally discovered PitNETs ranges from 20.6 to 24.6% [57,63,84,85]. Among these, hypogonadism is the most common [85]. When hypopituitarism occurs during follow-up, 60% of cases are not associated with tumor expansion [84]. A report also examined the endocrine function of clinically nonfunctioning PitNETs discovered incidentally and operated on compared to symptomatic nonfunctioning PitNETs [58]. In the incidentaloma group, 37.7% had severe GH deficiency, and 42.9% had some form of hypopituitarism. However, it should be noted that this study was conducted in surgical cases and PitNETs is a rather large incidental tumor.

Finally, if family history suggests multiple endocrine neoplasia syndrome, screening tests and follow-up should be performed according to the suspected syndrome [7].

There is also debate about the extent to which actual endocrinological testing should be performed, although major guidelines recommend only an initial assessment of hormones. The assessment of hypersecretion should include the measurement of prolactin, GH, and adrenocorticotropic hormone (ACTH) levels among anterior pituitary hormones [7]. Conversely, there are many negative opinions on hormone load tests [7]. In particular, hormone load tests for large tumors with suprasellar extension of the sella turcica should be performed with caution due to the risk of pituitary apoplexy [86]. Pituitary apoplexy associated with hormone load testing is more common in clinically nonfunctioning PitNETs, and few incidences are confined to the sella turcica. It has also been reported that the frequency of pituitary apoplexy after hormone loading tests is higher in thyroid-releasing hormone loading tests [87].

Toini et al. found that Cushing’s disease can be systematically screened in patients with pituitary incidentaloma by performing urinary free cortisol, serum or salivary cortisol at night, in addition to a 1 mg dexamethasone suppression test. The results showed pituitary hypercortisolemia in 7.3% of the patients, and ACTH positivity in 4.4% of the pathological findings [88]. However, the study refuted this, stating that tests to detect Cushing’s disease have a high false-positive rate and that careful consideration should be given to the balance of benefits and risks of performing pituitary incidentaloma surgery to treat subclinical hypercortisolemia [89]. In summary, it is advisable to measure early morning fasting basal values of GH, prolactin, ACTH, luteinizing hormone, follicle-stimulating hormone, TSH, free thyroxine, cortisol, testosterone (males), and estradiol (females) at the initial evaluation and at follow-up.

Regarding imaging studies, if only CT has been performed, it is recommended to add an MRI, especially a detailed MRI of the thin slice with a focus on the sella turcica [7]. In this case, coronal and sagittal views focused on the pituitary gland are necessary. T2-weighted images are also important, and contrast-enhanced MRI is recommended if possible. In cases of pituitary incidentaloma, when the tumor is in contact with the optic nerve during imaging, visual function should be evaluated, including visual field testing, even in the absence of subjective symptoms. This is because visual field defects may be detected by visual field testing even in the absence of symptoms. Visual field abnormalities have been reported in 4.5–11.1% of pituitary incidentalomas [1,3]. It has also been reported that 100% of patients with a suprasellar volume greater than 1.5 mL have visual field defects [90]. In France, the guidelines state that visual function evaluation is not always necessary for microadenomas or small tumors far from the optic chiasm [74].

### 5.2. Natural History

The natural history of pituitary incidentalomas differs from that of PitNETs and other cystic lesions, such as Rathke cleft cysts. There are numerous reports on the natural history of PitNETs treated without surgical or medical treatment [1,2,3,4,5,6,53,55,84,85,91,92,93]. According to them, micro PitNETs have an increase rate of 0–40%, while macro PitNETs have an increase rate of approximately 7–51%, and thus macro PitNETs are naturally more common (Table 4). Arita et al. also reported the risk of pituitary stroke during the course of the disease [2].

There have been several reports on the natural history of cystic lesions, mainly Rathke cleft cysts. Igarashi et al. reported that 30% of 10 cystic lesions remained unchanged, 40% were reduced, and 30% were enlarged after reduction [94]. Sanno et al. also reported that when the presumptive diagnosis was Rathke cleft cyst, only 5/94 (5.3%) were enlarged during the course of the disease [5]. This indicates that cystic lesions rarely enlarge and sometimes shrink. A meta-analysis summarizing these results also found that the annual enlargement rate was 12.5% for macro-PitNETs and 5.7% for solid lesions, while it was 3.3% for micro-PitNETs and 0.5% for cystic lesions, which tended to be lower. The worsening of visual dysfunction due to pituitary apoplexy is also rare [93]. However, it is important to note that although the opportunity for discovery is coincidental, cases of visual dysfunction and hypopituitarism were included in the detailed examination. In addition, cases in which surgery was avoided at the patient’s request were also included. Therefore, caution should be exercised when interpreting these cases.

## 6. Surgical Indications

As we have discussed, pituitary incidentalomas include different types of lesions. Therefore, there are many things that are not known about the follow-up of pituitary incidentalomas.

In this section, we discuss the recommendations for follow-up by the American Endocrine Society [7]. Consequently, surgery is recommended in cases in which tumor compression causes external ophthalmoplegia and visual field disturbances. Surgery is also indicated in functioning PitNETs other than lactotroph PitNETs, and in cases of pituitary apoplexy with visual dysfunction. In addition, MRI showing the lesion in contact with or compressing the optic nerve or optic chiasm is also an indication for surgery. The guidelines also suggest surgery, albeit as a weak recommendation, in the following cases. Surgery is suggested when the lesion is clearly enlarged, when endocrine dysfunction is present, or when there are unremitting headaches. However, it should be noted that it is often difficult to determine whether the headache is caused by a pituitary lesion. Anti-CGRP monoclonal antibodies are extremely effective, especially if the headache is due to migraine. In addition, surgery may also be suggested if the patient has a lesion near the optic chiasm and is planning a pregnancy. This is due to the physiologic pituitary enlargement that occurs during pregnancy, which carries the risk of tumor compression in the optic nerve or optic chiasm. However, there is no clear evidence for this, and caution should be exercised in its interpretation.

The French guidelines also describe surgical indications [74]. This is similar to the indication for surgery by the American Endocrine Society, but with some differences. According to this, surgery should be performed for uncooperative patients who escape regular follow-up, and male patients with macro PitNETs close to the optic nerve receiving anticoagulants. Pituitary apoplexy is more common in male patients. Anticoagulation therapy is often a predisposing factor, but some reports indicate that diabetes and hypertension are unrelated [95]. They also stated that surgery should be selected for cases that need to be differentiated from malignancy. Although malignant tumors are rarely asymptomatic, patients with suspected germ cell tumors or pituitary metastases should be carefully monitored.

## 7. Follow-Up

The American Association of Endocrinologists guidelines also describe follow-up procedures [7]. First, MRI should be performed once a year for 3 years for pituitary incidentalomas less than 1 cm in size. Thereafter, if there is no change in the size of the lesion, the follow-up frequency should be reduced. Nevertheless, for lesions larger than 1 cm, an MRI is performed every six months, and visual function and endocrine evaluations are performed simultaneously. After three years, follow-up should be performed once a year and, thereafter, the frequency should be reduced if there is no change in the size of the lesion. The French guidelines differ regarding follow-up for small lesions [74]. That is, lesions less than 5 mm in diameter should not be followed up at all, while lesions with a diameter of 6–9 mm should be followed up for 2 years. In addition, guidelines from the Department of Neuroradiology have different guidelines for the follow-up of macro- and micro-PitNETs diagnosed based on imaging findings and their nonfunctioning endocrine status. For micro-PitNETs, MRI surveillance is performed after 1 year, followed by MRI every 1–2 years, or less frequently if the disease is stable. In the case of macro PitNETs, if there is no visual dysfunction or abnormal pituitary function, MRI is performed after the first 6 months, then annually for the next 5 years, and less often if the disease is stable [81]. Finally, our proposed scheme for the management of the pituitary incidentalomas is shown in Figure 1.

## 8. Treatment Outcome (Including Endocrine Function)

As mentioned above, surgical treatment is the treatment of choice for pituitary incidentalomas when surgery is indicated after careful consultation with the patient. In cases where excessive hormone secretion is observed and a lactotroph PitNET is diagnosed, dopamine agonist treatment is the treatment of choice [7]. In addition, if surgery is performed, it should be performed in a high-volume center. This is because the incidence of surgical complications is higher in patients with less surgical experience [96]. In addition, regarding surgical technique, transnasal endoscopic surgery is recommended, except in cases of irregular tumor extension [74].

There are also scattered reports that incidental PitNETs that require surgery have a better prognosis than symptomatic PitNETs. Morinaga et al. compared the outcomes of patients with pituitary incidentalomas requiring surgery and those with symptomatic PitNETs according to the American Endocrine Society guidelines [97]. They found significantly less postoperative hormone dysfunction, fewer residual tumor sizes, and less need for reoperation in the pituitary incidentaloma group. Ono et al. also evaluated postoperative endocrine function and found that pituitary function was better preserved in the pituitary incidentaloma group than in the symptomatic group. In a multivariate analysis, the factors associated with severe hypopituitarism of three or more systems after surgery were male sex, maximum tumor diameter, and symptomatic tumor [58]. Additionally, Losa et al. examined surgical results among patients with pituitary incidentaloma, dividing them into two groups: those who were symptomatic and those who were completely asymptomatic, with visual function and endocrine examination as additional tests [8]. According to the results, postoperative tumor residuals were 31.2% and 8.9% in the symptomatic and asymptomatic groups, respectively. The 5-year recurrence-free rates in the symptomatic and asymptomatic groups were 77.9% and 86.8%, respectively. In other words, among patients with pituitary incidentalomas, the truly asymptomatic group had better results. Messerer et al. similarly divided incidentally discovered nonfunctioning pituitary incidentalomas into symptomatic and asymptomatic groups [98]. The total removal rate was higher in the asymptomatic group and strongly associated with the Knosp classification. Postoperative rates of visual dysfunction and endocrine disturbances were also higher in the symptomatic group. Seltzer et al. also reported the long-term results of incidentally discovered PitNETs [99]. They found that at an average follow-up of 61 months, 50% of the patients had improved headache and 54.5% had improved visual function. Furthermore, 25% of the patients showed improvement in endocrine function. These results suggest that treatment should be tailored to younger patients at risk for endocrine or visual dysfunction.

The results of tumor pathology provide important information for postoperative follow-up. Suzuki et al. reported that when comparing pituitary incidentalomas and symptomatic clinically nonfunctioning PitNETs, the MIB-1 index was significantly lower in pituitary incidentalomas [100]. However, it is important to note the aggressive nature of certain pathological types of PitNETs. Specifically, the immature pluripotent PitNET lineage PIT1 [101], the Crooke’s cell tumor variant corticotroph PitNET [102,103], null cell PitNETs in which all pituitary transcription factors and adenohypophyseal hormones are negative [104,105,106], and nonfunctioning biochemically silent corticotroph PitNETs [107,108]. Of these, null cell PitNETs and biochemically nonfunctional silent corticotroph PitNETs should be noted because they are endocrinologically nonfunctional and may be included in the category of pituitary incidentalomas.

## 9. Conclusions

With advances in imaging and other types of tests, the opportunities for exposure to pituitary incidentalomas are expected to continue to increase. Specifically, follow-up based on the revised WHO classification of endocrine and neuroendocrine tumors is important. However, the costs of evaluating these patients should also be considered. Randall et al. reported that the cost of evaluating pituitary incidentalomas was USD 7 million in the United States in 2005 [109]. In the future, pituitary incidentalomas could be managed when a balance between cost and benefit is considered. Therefore, studies with strong evidence are required to consider treatment guidelines for pituitary incidentalomas.

## Figures and Tables

**Figure 1 cancers-14-04324-f001:**
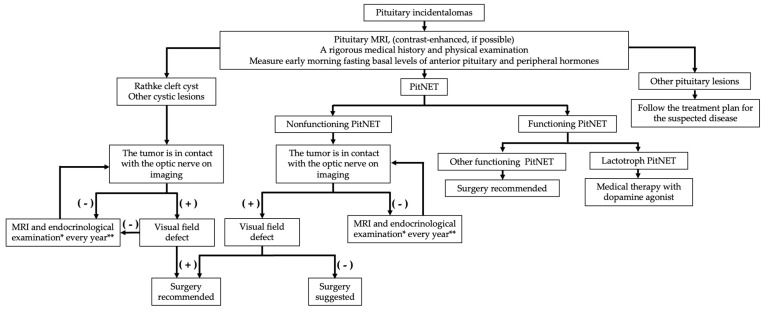
Our proposed scheme for the management of the pituitary incidentalomas. PitNET: pituitary neuroendocrine tumor. * Suggest surgery if there is progressive hypopituitarism. ** The first follow-up will be performed in half a year.

**Table 1 cancers-14-04324-t001:** Frequency of PitNETs found at autopsy.

Year	Authors	Number of Pituitaries Studied	Number of PitNETs Found	Frequency (%)
1936	Costello [11]	1000	225	22.5
1959	Sommers [12]	400	26	6.5
1969	Hardy [13]	1000	27	2.7
1971	McCormick [14]	1600	140	8.8
1973	Haugen [15]	170	33	19.4
1980	Kovacs [16]	152	20	13.2
1981	Burrow [17]	120	32	26.7
1981	Max [18]	500	9	1.8
1981	Muhr [19]	205	3	1.5
1981	Parent [20]	500	42	8.4
1982	Chambers [21]	100	14	14.0
1982	Schwesinger [22]	5100	485	9.5
1983	Coulon [23]	100	10	10.0
1984	Siqueira [24]	450	39	9.5
1991	Kontogeorgos [25]	470	49	10.4
1992	Marin [26]	210	35	16.7
1994	Teramoto [27]	1000	51	5.1
1995	Camaris [28]	423	14	3.2
1999	Tomita [29]	100	24	24.0
2001	Kurosaki [30]	692	79	11.4
2006	Buurman [32]	3048	334	11.0
2007	Furgal-Borzych [33]	151	47	31.1
2007	Kim [34]	120	7	5.8
2007	Rittierodt [35]	228	7	3.0
2011	Aghakhani [36]	485	61	12.6

PitNET: pituitary neuroendocrine tumor.

**Table 2 cancers-14-04324-t002:** Frequency of PIs detected during imaging.

Year	Authors	Subject	Age (y/o)	Number of Pituitaries Studied	Number of PIs Found	Frequency (%)	Remarks
**Examination using the CT**							
1982	Chambers [21]	Patients with orbital symptoms	NC	50	10	20.0	Contrast-enhanced high-resolution CT
1997	Nammour [39]	Consecutive patients undergoing head CT at a single institution	57	3550	7	0.2	Macro PitNETs only
2012	Pette [43]	Patients examined for dental implant therapy	10–91 Ave. 64.73	318	2	0.63	Cone beam CT
**Examination using the MRI**							
1994	Chong [37]	Healthy adult volunteers	22–68 Ave. 34	52	20	38.5	Local low signal on T1-weighted image
1994	Hall [38]	Healthy adult volunteers	18–60	100	10	10.0	Contrast-enhanced MRI of the pituitary gland
1997	Yue [40]	Patients with cardiovascular and cerebrovascular disease	≥65	3672	6	0.16	
2007	Vernooij [41]	Health checkups for residents near Rotterdam, the Netherlands	45.7–96.7 Ave. 63.3	2000	6	0.3	Macro PitNETs only
2013	Sandeman [44]	Residents of Edinburgh	Ave. 72.5	700	2	0.28	
2013	Hegenscheid [45]	Residents of northwest Germany	21–88 Ave. 53	2500	9	0.36	
2016	Bos [46]	Residents of the Netherlands over 45 y/o	Ave. 64.9	5800	67	1.2	
2016	Håberg [47]	Healthy adult volunteers	50–66	1006	3	0.3	
2017	Boutet [48]	French retirees over 65 y/o	Ave. 75.3	503	11	2.2	
2021	Yoo [50]	Patients without endocrine abnormalities	<18	365	76	20.8	Contrast-enhanced MRI
2022	Lohner [51]	Residents of Bonn, Germany	55	3589	3	0.08	Using 3Tesla MRI, macro PitNETs only
Examination using FDG PET/CT							
2010	Jeong [42]	Patients with malignancy or screened for cancer	NC	40,967	30	0.073	

PI: pituitary incidentaloma; CT: computed tomography; MRI: magnetic resonance imaging; FDG PET: 18F-fluorodeoxyglucose positron emission tomography; y/o: years old; NC: not clear; Ave.: average.

**Table 3 cancers-14-04324-t003:** Typical neoplastic or mass lesions in and around the pituitary gland.

Neoplastic Lesion
PitNET, craniopharyngioma, pituicyte tumor, meningioma, chordoma, neuroblastoma, germ cell tumor, lymphoma, metastatic tumor, Langerhans cell histiocytosis
**Cystic lesion**
Rathke’s cleft cyst, arachnoid cyst
**Non-neoplastic lesion**
Hypophysitis, IgG4-related disease, sarcoidosis, granulomatosis with polyangiitis, infective granuloma (tuberculosis, fungus, bacterial), abscess, pituitary hyperplasia, empty sella, cerebral aneurysm

PitNET: pituitary neuroendocrine tumor; IgG4: immunoglobulin G4.

**Table 4 cancers-14-04324-t004:** Tumor size changes in pituitary incidentalomas with an estimated diagnosis of PitNETs.

Year	Authors	Total	Increased (%)	Decreased (%)	No Change (%)	Follow-Up Period (Months)
Micro PitNETs						
1990	Reincke [3]	7	1 (14.3)	1 (14.3)	5 (71.4)	22
1995	Donovan [4]	15	0	4 (26.7)	11 (73.3)	76.8
1999	Feldkamp [1]	31	1 (3.2)	1 (3.2)	29 (93.6)	32.4
2003	Sanno [5]	74	10 (13.5)	7 (9.5)	57 (77)	27
2004	Day [6]	11	1 (9.1)	0	10 (90.9)	38.4
2006	Arita [2]	5	2 (40)	0	3 (60)	61.9
2007	Karavitaki [92]	16	2 (12.5)	1(6.3)	13 (81.2)	42
2011	Anagnostis [53]	6	0	1 (16.7)	5 (83.3)	48
2020	Tresoldi [84]	132	12 (9.1)	28 (21.2)	92 (69.7)	36
Macro PitNETs						
1990	Reincke [3]	7	2 (28.6)	0	5 (71.4)	22
1995	Donovan [4]	16	5 (31.3)	0	11 (68.7)	76.8
1998	Nishizawa [91]	28	2 (7.1)	0	26 (92.9)	67.2
1999	Feldkamp [1]	19	5 (26.3)	1 (5.3)	13 (68.4)	32.4
2003	Sanno [5]	165	20 (12.1)	22 (13.3)	123 (74.6)	27
2004	Day [6]	7	1 (14.3)	0	6 (85.7)	38.4
2006	Arita [2]	37	19 (51.4)	0	18 (48.6)	61.9
2007	Karavitaki [92]	24	12 (50)	4 (16.7)	8 (33.3)	42
2011	Anagnostis [53]	3	1 (33.3)	0	2 (66.7)	48
2020	Tresoldi [84]	71	19 (26.8)	4 (5.6)	48 (67.6)	36
No distinction between the micro PitNET and the macro PitNET						
2016	Imran [55]	113	21 (18.6)	2 (1.8)	90 (79.6)	36
2017	Iglesias [85]	26	1 (3.8)	2 (7.7)	23 (88.5)	15.5

PitNET: pituitary neuroendocrine tumor.

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
