# Peer review of "An Overview of Pituitary Incidentalomas: Diagnosis, Clinical Features, and Management"

_cancers, 2022, doi:10.3390/cancers14174324_

Round 1
Reviewer 1 Report
I feel the authors are well-done with literature search to describe pituitary incidentaloma. The article has almost enough information outlining pituitary incidentalomas. However, there are a lot of tautology parts and duplications of the information that is already on the tables. Besides, when referencing data from previous literature, I do not think the authors’ names need to be displayed; rather, aiming for conciseness, summarizing data is recommended. For example, most of the part from line 90 through line 118 can be removed, because the same information is on Table 2 in a more informative way. Wording and grammar need to be refined.
Line 16-17. “Majority of pituitary incidentalomas are pituitary neuroendocrine tumors (PitNETs)” “PitNET, which accounts for majority of pituitary incidentalomas”
Please consider correcting this tautology.
Line 77. “On the contrary, a meta-analysis by Ezzat et al. found an estimated prevalence of 14.4 tumors in autopsy cases of 14.4%” Typo? Please correct.
Line 82. “They represented 6.1% of the total, with PitNET and hyperplasia at 2% and Rathke cleft cysts at 3.7%” I do not understand what this sentence means. Please explain?
Line 90-95. The same information can be obtained from Table 2, and thus this section does not bring any new information. Please consider modification or simply remove it.
Epidemiology is quite important for incidental tumors. In particular, Table 2 is informative but can be refined to be more informative. (A) Age (mean, median, or inclusion criterion) is an important factor and thus should be added in the column. (B) Were examinees all healthy volunteers or those who underwent MRI for any other reasons? Please also add this information to Table 2, creating a new column titled “subject”. Moreover, by doing so, I think most of the parts between line 96 and 112 can be shortened, because this part appears to be just a duplication of the information described in Table 2. Last, please state your own summary of the frequency of pituitary incidentalomas by synthesizing the results of many studies outlined in the article. What number do you think represents the frequency appropriately? This will be more useful to readers, instead of just a constellation of the known facts.
Line 160. “However, this study was based on the WHO classification”
Why this sentence starts with “however”? Because this was based on a “previous” version?
Line 180-181. The authors show the prevalence of PitNETs in several countries (Table 4). This information is fine, but how come this is placed in the subsection of “Evaluation at the time of detection”? Besides, this Table is not for incidentalomas but for PitNET, and thus can be somewhat misleading and confusing? It is of course of importance to rule out functioning PitNETs, but the significance of ruling out functioning PitNETs could be explained without Table 4.
Line 186-213. This part is a bit too long. Please shorten this part significantly, because, in summary, I think these paragraphs essentially tell the importance of endocinological evaluation when finding pituitary indcidentalomas. The authors do not need to constellate these many literature data to support its importance.
Line 225. “The frequency of TRH loading tests was also high” This sentence is a bit difficult to understand. Perhaps the authors intended to mean “the risk of pituitary apoplexy following/after TRH loading was higher than the other hormone load tests.” Please consider rephrasing.
Please more clearly state what types of hormone tests are required as the initial and follow-up assessments? This information is more important to readers than the constellation of the previous data.
Please state what kinds of MRI sequences should be used at the initial assessment and follow-up studies.
Line 238-242. Please consolidate the numbers and shorten the sentences. Detailed authors’ names do not need to be described.
Line 237-244. Empty sella and pituitary hyperplasia need to be added to Table 3 for integrity. Also, please reconsider where this information should be placed. I think this is better placed in the differential diagnosis section.
Line 245 and 247. “In cases of pituitary incidentaloma” was used twice back-to-back.
Please avoid tautology.
Line 249-254. Again, please consolidate the numbers and shorten the sentences. Detailed authors’ names do not need to be described.
Line 258-
The authors mentioned that the natural history of pituitary incidentaloma differs from PitNETs and other cystic lesions, but the following sentences are constellations of data on natural histories of PitNET and Rathke’s cleft cyst. This means, this section essentially does not have any significant message in terms of the natural history of incidentalomas? Nevertheless, in the top of the next paragraph, the authors described that based on the epidemiology and natural history of pituitary incidentalomas, the American Endocrine Society provides a commentary on how to follow up, as if the natural history of pituitary incidentalomas have been well-outlined. These sentences are inconsistent; please clarify.
Line 280-
I feel this section mainly focuses on surgical indication, not follow-up. Please consider changing the section title. Also, describing the follow-up protocol is quite important for this topic; thus, separating these two is recommended.
Line 282- “Consequently, surgery is recommended in cases of visual field defects or abnormal visual function other than those due to tumor compression (e.g., external ophthalmoplegia).”
This sentence is unclear. What is the “abnormal visual function other than those due to tumor compression?” The authors provided external ophthalmoplegia as one example; however, how can external ophthalmoplegia occur without tumor compression?
Line 284- “Furthermore, the lesion contacts or compresses the optic nerve or optic chiasm on MRI.”
Typo?
Line 287. I do not think “However” is appropriate given the surrounding context?
Line 289- “when the patient has a lesion near the optic chiasm and is planning a pregnancy”
Does this mean that the authors always recommend surgery for female patients before pregnancy who have pituitary incidentalomas that are just close to but not in contact with the optic chiasm? Is there any evidence that strongly advocate surgical resection for such a situation?
Line 290. “when the patient presents with unremitting headaches.”
Please tone down here. This is not an absolute surgical indication. Besides, recent advances on anti-CGRP antibodies may resolve some of them.
Line 294- “This is because of the risk of pituitary apoplexy is higher in men and patients receiving anticoagulation therapy than in cardiovascular risk factors such as hypertension or diabetes mellitus”
Please consider modification of the wording. Even though I can catch the meaning, this sentence is not grammatically correct.
Adding a scheme that illustrates flow of surgical indications and follow-up protocol would be quite helpful and can be a main theme of this paper. Please consider.
Author Response
Comments and Suggestions for Authors
I feel the authors are well-done with literature search to describe pituitary incidentaloma. The article has almost enough information outlining pituitary incidentalomas. However, there are a lot of tautology parts and duplications of the information that is already on the tables. Besides, when referencing data from previous literature, I do not think the authors’ names need to be displayed; rather, aiming for conciseness, summarizing data is recommended. For example, most of the part from line 90 through line 118 can be removed, because the same information is on Table 2 in a more informative way. Wording and grammar need to be refined.
Thank you very much for your detailed remarks. We have reconsidered the tautology and grammar and made corrections. We have also made efforts to simplify the data.
---------------------------------------------------------------------------------
Line 16-17. “Majority of pituitary incidentalomas are pituitary neuroendocrine tumors (PitNETs)” “PitNET, which accounts for majority of pituitary incidentalomas”
Please consider correcting this tautology.
Thank you very much for pointing this out. We have deleted ''which accounts for the majority of pituitary incidentalomas'' on line 16.
---------------------------------------------------------------------------------
Line 77. “On the contrary, a meta-analysis by Ezzat et al. found an estimated prevalence of 14.4 tumors in autopsy cases of 14.4%” Typo? Please correct.
Thank you very much for pointing this out. We have deleted ''on the contrary'' on line 85 and restructured the sentence to accurately indicate the values.
---------------------------------------------------------------------------------
Line 82. “They represented 6.1% of the total, with PitNET and hyperplasia at 2% and Rathke cleft cysts at 3.7%” I do not understand what this sentence means. Please explain?
Thank you very much for your comment. It was indeed a very confusing sentence; we apologize for the lack of clarity. We have made the following revisions (lines 88–95):
‘’Teramoto et al. found a kind of incidental lesion in 17.8% of pituitary autopsy cases. The frequency of incidental pathological lesions larger than 2 mm was considered clinically significant, as they were detectable using MRI or other imaging studies. They comprised 6.1% of the total cases, with PitNET and hyperplasia accounting for 2%, and Rathke cleft cysts for 3.7% [27]. Moreover, they suggested that incidental lesions should be considered as a cause of false positive findings when imaging reveals functional pituitary microadenomas manifesting with Cushing's disease.’’
---------------------------------------------------------------------------------
Line 90-95. The same information can be obtained from Table 2, and thus this section does not bring any new information. Please consider modification or simply remove it.
Thank you very much for your comment. We have deleted this section per your suggestion.
---------------------------------------------------------------------------------
Epidemiology is quite important for incidental tumors. In particular, Table 2 is informative but can be refined to be more informative. (A) Age (mean, median, or inclusion criterion) is an important factor and thus should be added in the column. (B) Were examinees all healthy volunteers or those who underwent MRI for any other reasons? Please also add this information to Table 2, creating a new column titled “subject”. Moreover, by doing so, I think most of the parts between line 96 and 112 can be shortened, because this part appears to be just a duplication of the information described in Table 2. Last, please state your own summary of the frequency of pituitary incidentalomas by synthesizing the results of many studies outlined in the article. What number do you think represents the frequency appropriately? This will be more useful to readers, instead of just a constellation of the known facts.
Thank you very much for your important remarks. We have added columns for age and subjects in Table 2. We have also discussed the frequency of pituitary incidentalomas and synthesized the results of studies on the frequency of pituitary incidentalomas in autopsy cases and imaging studies (lines 169–175).
---------------------------------------------------------------------------------
Line 160. “However, this study was based on the WHO classification”
Why this sentence starts with “however”? Because this was based on a “previous” version?
Yes, we used “however” because it was based on the previous version. We have added some details as follows (lines 216–217):
‘’this study was based on the previous WHO Classification, 4th edition, which was revised in 2017.’’
---------------------------------------------------------------------------------
Line 180-181. The authors show the prevalence of PitNETs in several countries (Table 4). This information is fine, but how come this is placed in the subsection of “Evaluation at the time of detection”? Besides, this Table is not for incidentalomas but for PitNET, and thus can be somewhat misleading and confusing? It is of course of importance to rule out functioning PitNETs, but the significance of ruling out functioning PitNETs could be explained without Table 4.
Thank you very much for your suggestion. We have deleted Table 4. We have also moved the relevant information to the “Type of pituitary incidentalomas” section (lines 222–229).
---------------------------------------------------------------------------------
Line 186-213. This part is a bit too long. Please shorten this part significantly, because, in summary, I think these paragraphs essentially tell the importance of endocinological evaluation when finding pituitary indcidentalomas. The authors do not need to constellate these many literature data to support its importance.
Thank you very much for your suggestion. We have moved the relevant information to another section (lines 222–229). We have also significantly shortened the text that follows.
---------------------------------------------------------------------------------
Line 225. “The frequency of TRH loading tests was also high” This sentence is a bit difficult to understand. Perhaps the authors intended to mean “the risk of pituitary apoplexy following/after TRH loading was higher than the other hormone load tests.” Please consider rephrasing.
Thank you very much for your comment. Per your suggestion, we have corrected this part as follows (lines 344–346):
'' It has also been reported that the frequency of pituitary apoplexy after hormone loading tests is higher in thyroid releasing hormone loading tests.''
---------------------------------------------------------------------------------
Please more clearly state what types of hormone tests are required as the initial and follow-up assessments? This information is more important to readers than the constellation of the previous data.
Thank you for your suggestion. We have added the following information on lines 354–357:
‘’In summary, it is advisable to measure early morning fasting basal values of GH, prolactin, ACTH, luteinizing hormone, follicle-stimulating hormone, TSH, free thyroxine, cortisol, testosterone (males), and estradiol (females) at the initial evaluation and follow-up.’’
---------------------------------------------------------------------------------
Please state what kinds of MRI sequences should be used at the initial assessment and follow-up studies.
Thank you for this suggested. We have added the MRI sequences that should be used on lines 359–362.
---------------------------------------------------------------------------------
Line 238-242. Please consolidate the numbers and shorten the sentences. Detailed authors’ names do not need to be described.
Thank you very much for your important remarks. This section has been moved to lines 230–236. The sentences have also been reorganized.
---------------------------------------------------------------------------------
Line 237-244. Empty sella and pituitary hyperplasia need to be added to Table 3 for integrity. Also, please reconsider where this information should be placed. I think this is better placed in the differential diagnosis section.
Thank you very much for your suggestion. We have added pituitary hyperplasia and empty sella to the non-neoplastic lesions section in Table 3. We have moved this statement to lines 230–238.
---------------------------------------------------------------------------------
Line 245 and 247. “In cases of pituitary incidentaloma” was used twice back-to-back.
Please avoid tautology.
Thank you for your suggestion. We have revised this section as shown below (lines 364–365).
‘’This is because visual field defects may be detected by visual field testing even in the absence of symptoms.’’
---------------------------------------------------------------------------------
Line 249-254. Again, please consolidate the numbers and shorten the sentences. Detailed authors’ names do not need to be described.
Thank you very much for your suggestion. We have shortened the sentences in this section (lines 365–368).
---------------------------------------------------------------------------------
Line 258-
The authors mentioned that the natural history of pituitary incidentaloma differs from PitNETs and other cystic lesions, but the following sentences are constellations of data on natural histories of PitNET and Rathke’s cleft cyst. This means, this section essentially does not have any significant message in terms of the natural history of incidentalomas? Nevertheless, in the top of the next paragraph, the authors described that based on the epidemiology and natural history of pituitary incidentalomas, the American Endocrine Society provides a commentary on how to follow up, as if the natural history of pituitary incidentalomas have been well-outlined. These sentences are inconsistent; please clarify.
We agree that the information was incoherent and confusing; we apologize for this.
We have deleted the phrase “Based on the epidemiology and natural history of pituitary incidentalomas” on line 281. We have added the following sentence: “As we have discussed, pituitary incidentalomas include the different types of lesions. Therefore, there are many things that are not known about the follow-up of pituitary incidentalomas” (lines 456–458).
---------------------------------------------------------------------------------
Line 280-
I feel this section mainly focuses on surgical indication, not follow-up. Please consider changing the section title. Also, describing the follow-up protocol is quite important for this topic; thus, separating these two is recommended.
Thank you for your important remarks. As you indicated, we have made the section from lines 455–523 surgical indications.
---------------------------------------------------------------------------------
Line 282- “Consequently, surgery is recommended in cases of visual field defects or abnormal visual function other than those due to tumor compression (e.g., external ophthalmoplegia).”
This sentence is unclear. What is the “abnormal visual function other than those due to tumor compression?” The authors provided external ophthalmoplegia as one example; however, how can external ophthalmoplegia occur without tumor compression?
The description was incorrect and confusing; we apologize for the inconvenience. We have changed these descriptions as follows (lines 460–464):
‘’Consequently, surgery is recommended in cases in which tumor compression causes external ophthalmoplegia and visual field disturbances. Surgery is also indicated in functioning PitNETs other than lactotroph PitNETs and in cases of pituitary apoplexy with visual dysfunction. In addition, MRI showing the lesion in contact with or compressing the optic nerve or optic chiasm is also an indication for surgery.’’
---------------------------------------------------------------------------------
Line 284- “Furthermore, the lesion contacts or compresses the optic nerve or optic chiasm on MRI.”
Typo?
The description was incorrect and confusing; we apologize for the inconvenience. Per your comment, we have made the following revision:
‘’Consequently, surgery is recommended in cases in which tumor compression causes external ophthalmoplegia and visual field disturbances. Surgery is also indicated in functioning PitNETs other than lactotroph PitNETs and in cases of pituitary apoplexy with visual dysfunction. In addition, MRI showing the lesion in contact with or compressing the optic nerve or optic chiasm is also an indication for surgery (lines 460 - 464).’’
---------------------------------------------------------------------------------
Line 287. I do not think “However” is appropriate given the surrounding context?
You are correct. We have deleted “However.”
---------------------------------------------------------------------------------
Line 289- “when the patient has a lesion near the optic chiasm and is planning a pregnancy”
Does this mean that the authors always recommend surgery for female patients before pregnancy who have pituitary incidentalomas that are just close to but not in contact with the optic chiasm? Is there any evidence that strongly advocate surgical resection for such a situation?
Thank you very much for your questions. The US guidelines suggest surgery (weak recommendation) for women who wish to become pregnant and have an incidental pituitary tumor in close proximity to the optic chiasm. This is stated to be due to the physiologic enlargement of the pituitary gland during pregnancy, but there does not seem to be strong evidence for this. Therefore, this section has been corrected to weak recommendation (lines 509–513).
-------------------------------------------------------------------------------------------------------
Line 290. “when the patient presents with unremitting headaches.”
Please tone down here. This is not an absolute surgical indication. Besides, recent advances on anti-CGRP antibodies may resolve some of them.
Thank you very much for pointing this out. This part has also been changed to suggest surgery (weak recommendation) and corrected (lines 464–509).
---------------------------------------------------------------------------------
Line 294- “This is because of the risk of pituitary apoplexy is higher in men and patients receiving anticoagulation therapy than in cardiovascular risk factors such as hypertension or diabetes mellitus”
Please consider modification of the wording. Even though I can catch the meaning, this sentence is not grammatically correct.
Thank you very much for this remark. We have corrected this part as follows (lines 518–520)
‘’Pituitary apoplexy is more common in male patients. Anticoagulation therapy is often a predisposing factor, but some reports indicate that diabetes and hypertension are unrelated.’’
-------------------------------------------------------------------------------------------------------
Adding a scheme that illustrates flow of surgical indications and follow-up protocol would be quite helpful and can be a main theme of this paper. Please consider.
Thank you very much for your attention. Our proposed scheme for the management of the pituitary incidentalomas is shown in Figure 1 and lines 542–543 of the text.
---------------------------------------------------------------------------------
I have corrected the paper according to the reviewers' comments and had it proofread by a native English speaker. Finally, I enclose a certificate of proofreading in English.
Thank you again for your valuable comments. We request that our paper be reconsidered for publication in the Cancers.
Reviewer 2 Report
The authors provide a nice, rich, and detailed overview about an interesting topic with the adequate support of clear tables where needed. I do agree with the conclusive message about the right balance between costs and benefits of treating pituitary incidentalomas. After a careful revision I would consider the paper for publication after a minor language check from a native English speaker.
Author Response
Comments and Suggestions for Authors
The authors provide a nice, rich, and detailed overview about an interesting topic with the adequate support of clear tables where needed. I do agree with the conclusive message about the right balance between costs and benefits of treating pituitary incidentalomas. After a careful revision I would consider the paper for publication after a minor language check from a native English speaker.
Thank you very much for your valuable comments. I have corrected the paper according to the other reviewers' comments and had it proofread by a native English speaker. Finally, I enclose a certificate of proofreading in English.
Thank you again for your valuable comments. We request that our paper be reconsidered for publication in the Cancers.
Round 2
Reviewer 1 Report
The authors have responded well to my suggestions. Thank you very much. My last concern is that the manuscript still looks a bit too long. If the editor redommends the authors to shorten it, please do so. Otherwise I can accept the manuscript as is.